# The Seamless Solar Radiation (SESORA) Forecast for Solar Surface Irradiance—Method and Validation

**Isabel Urbich** [1,*] [iD], **Jörg Bendix** [2] [iD] and **Richard Müller** [1] [iD]

1   Department for Research and Development, Deutscher Wetterdienst, Frankfurter Straße 135, 63067 Offenbach, Germany; richard.mueller@dwd.de
2   Faculty of Geography, Philipps-Universität Marburg, Deutschhausstraße 12, 35032 Marburg, Germany; bendix@staff.uni-marburg.de
*   Correspondence: isabel.urbich@dwd.de; Tel.: +49-(0)69-8062-2475

**Abstract:** Due to the integration of fluctuating weather-dependent energy sources into the grid, the importance of weather and power forecasts grows constantly. This paper describes the implementation of a short-term forecast of solar surface irradiance named SESORA (seamless solar radiation). It is based on the the optical flow of effective cloud albedo and available for Germany and parts of Europe. After the clouds are shifted by applying cloud motion vectors, solar radiation is calculated with SPECMAGIC NOW (Spectrally Resolved Mesoscale Atmospheric Global Irradiance Code), which computes the global irradiation spectrally resolved from satellite imagery. Due to the high spatial and temporal resolution of satellite measurements, solar radiation can be forecasted from 15 min up to 4 h or more with a spatial resolution of $0.05°$. An extensive validation of this short-term forecast is presented in this study containing two different validations based on either area or stations. The results are very promising as the mean RMSE (Root Mean Square Error) of this study equals 59 W/m$^2$ (absolute bias = 42 W/m$^2$) after 15 min, reaches its maximum of 142 W/m$^2$ (absolute bias = 97 W/m$^2$) after 165 min, and slowly decreases after that due to the setting of the sun. After a brief description of the method itself and the method of the validation the results will be presented and discussed.

**Keywords:** solar surface irradiance; specmagic; optical flow; renewable energies; pyranometer; nowcasting

## 1. Introduction

Over recent decades the overall need for an accurate spatiotemporal nowcasting of weather has increased due to the rising importance of renewable energies and the fluctuating energy supply due to the short-term variation in the governing atmospheric elements (e.g., clouds and solar radiation) [1–3]. Particularly if renewable energies are integrated into the grid it is very important to correctly forecast the weather, as well as power needs, to prevent grid instabilities. Instabilities may occur as solar energy and wind energy have a major impact on the load flows and for this reason, forecasts have to become more precise, especially in the short-term range of 0–4 h [4–7]. A merge between the numerical weather prediction (NWP) model and nowcasting will deliver a seamless product of the highest quality at any time. The main cause for instabilities in Germany, for instance, is the inhomogeneous distribution of wind turbines and photovoltaic systems, which often leads to capacity overloads. As a consequence, a forecast for solar irradiation as the basis of solar energy based on observations, also called nowcasting, will be developed in this work. As has been shown before, nowcasting with satellite data delivers better results for the first few hours compared to numerical weather prediction (NWP) models, due to its higher temporal and spatial resolution [8–10]. Moreover, NWP model runs usually need 3–6 h

of computation time, depending on the model, as a consequence of the data assimilation. Satellite measurements and derived nowcasting products, on the other hand, are available in near real time with a spatial resolution of 0.05°.

A common and frequently used approach to forecast solar surface irradiation is a neural network [1,11]. However, it has not yet been shown that neural networks have a higher forecast quality than cloud motion vectors or optical flow methods [12]. Another successful method is the use of a semi-Lagrangian scheme to calculate the advection of a flow [13–15]. In contrast to a constant vector, as used in optical flow methods, the vectors for Lagrangian trajectories are iteratively determined for each time step to allow rotation of a flow [16]. Nevertheless, the differences between constant vectors and Lagrangian trajectories only stand out when there is rotation. Furthermore, especially at the beginning, differences are small and grow with increasing forecast time or distance [16]. Since we foresee a merge with an NWP model in our algorithm after about 4 h, differences will most likely stay small. Moreover, additional features, like the use of a history of satellite images instead of only two frames, or the enabling of curved trajectories where the past movement of a pixel will be taken into consideration, will be included in the algorithm of the SESORA forecast in the near future. Another great advantage of the optical flow method is that the algorithm of TV-$L^1$ (Method based on total variation in the regularization term and the $L^1$-norm in the data fidelity term) is open source and a large community is constantly working on improving this and other methods by OpenCV (Open Source Computer Vision) [17]. There are many successful solar radiation forecasts that have been published in recent years that use cloud tracking methods with either geostationary satellites [3,18–20], total sky imagery [21], or ground sensors [22]. One method of cloud tracking is to derive cloud motion vectors (CMV) from satellite imagery. The general use of cloud motion vectors for nowcasting is widespread and many approaches have been proposed so far. Their application is not limited to forecasts in the scope of energy meteorology. In Guillot et al. [23] cloud motion vectors were derived from satellite imagery and utilized to forecast cloud displacement over complex terrain. Velden et al. [24] used atmospheric motion vectors (AMV) to better forecast the track of tropical cyclones. These vectors are derived from the infrared channel of MSG (Meteosat Second Generation) and since they do not only display the motion of the top of clouds they are called AMVs. In the field of solar radiation forecasts the use of cloud motion vectors is very common [25–27]. Comparable studies from Gallucci et al. [28] or Sirch et al. [29] used cloud motion vectors from MSG/SEVIRI (Spinning Enhanced Visible and Infrared Imager) to forecast solar surface radiation for up to 2 h. Due to different error measures and different study areas, a direct comparison of the quality reported in both studies is not possible. However, despite these small differences in validation method, the results are comparable and reported uncertainties are in a similar order. Gallucci et al. presented an RMSE (Root Mean Square Error) of 147 W/m$^2$ over Italy after 2 h of forecasting and Sirch et al. have found a correlation of 0.7 between the forecast and the observation after 2 h over Europe. Both studies reported higher errors due to convective clouds, which is a common problem in the scope of motion vectors since the method is not able to consider the formation and dissipation of clouds [12,28].

The here presented solar radiation nowcasting is based on the optical flow of effective cloud albedo (CAL) [12]. CAL can be retrieved from the reflectivity measured in the visible channel of MSG and is therefore available every 15 min [30]. The biggest advantage of using CAL for the optical flow estimation is that CAL has a direct connection to the cloud transmission and thus to the cloud effect on the solar surface irradiance. Apart from the effective cloud albedo, none of the other input parameters are forecasted. The reason for this is that SPECMAGIC NOW (Spectrally Resolved Mesoscale Atmospheric Global Irradiance Code) uses the same clear sky input data as in the Heliosat method used in SARAH-2 (Surcae Solar Radiation Data Set - Heliosat). Forecasting only CAL therefore enables a clear separation of the errors induced by the CAL nowcasting since it represents the dominant error source for short-term fluctuations of solar irradiance in Central Europe. A list of further input parameters can be found below. For further information about the retrieval of CAL and the below listed input data the reader may refer to Müller et al. or Trentmann [30–34].

1. Aerosol is based on the Monitoring Atmospheric Composition and Climate Project (MACC) [35].
2. $H_2O$ is taken from the European Centre for Medium-Range Weather Forecasts (ECMWF) [36].
3. Surface albedo is based on 20 different land-use types originating from the NASA CERES/SARB (Clouds and the Earth's Radiant Energy System / Surface Atmospheric Radiation Budget) Surface Properties Project [37,38].

The optical flow requires two subsequent satellite images as input for the calculation of cloud motion vectors. The resulting motion vectors are then applied to the latter of these two images to extrapolate the observed clouds into the future. Additional information about the optical flow method and the effective cloud albedo nowcasting can be found in Section 2.2. After the propagation of the cloud is determined, solar surface radiation is calculated with SPECMAGIC NOW, which computes the global radiation, spectrally resolved from satellite images in the visible channel [39]. A detailed description of the algorithm will follow in Section 2.3.

The results of the CAL nowcasting were very promising, as the error measures of the forecast clearly showed [12]. The results were verified with satellite data from MSG for the area of Europe and the same error measures as in this publication were used (Section 2.4). A validation of the SESORA forecast is nevertheless necessary because of the integration of SPECMAGIC NOW. New features like the variability of the solar zenith angle and an all sky consideration may lead to other errors than in Urbich et al. [12]. Moreover, the errors of solar surface irradiance are very important for the application in the scope of PV (Photovoltaic) systems because the errors of solar irradiance and photovoltaic power have an almost linear relation [40,41]. This can be derived from the performance curve of photovoltaic systems considering that the errors grow proportional to their associated values. This linear relation is a huge advantage concerning error growth over the cubical relation between wind and power for wind turbines [42,43]. This information could significantly improve proper management of the grid loads.

## 2. Materials and Methods

The following section describes the validation data that were provided by the Satellite Application Facility on Climate Monitoring (CM SAF), the Baseline Solar Radiation Network (BSRN), and the German Weather Service. These data are now being used for the validation of the SESORA forecast. The methods of the optical flow and SPECMAGIC NOW will be presented and explained. The CAL data used for the here presented solar radiation nowcasting is one of the products of SPECMAGIC NOW and for further information the reader may refer to Urbich et al. [12]. Moreover, the utilized error measures of Sections 3.1 and 3.2 will be listed and described at the end of this section.

### 2.1. Validation Data

### 2.1.1. SARAH-2

For the area based validation of our SESORA forecast we used SARAH-2 data from the CM SAF. The solar surface irradiance data from the SARAH-2 data set is the latest CM SAF climate data record of surface radiation based on the geostationary Meteosat satellite series [44]. In addition to solar surface radiation, the SARAH-2 dataset offers other global and direct radiation parameters but without a forecast. SARAH-2 covers the area of $-65°$ to $+65°$ in latitude and longitude with a spatial resolution of $0.05°$ [44]. The quality of the SARAH-2 data set in reference to BSRN data is well documented in Pfeifroth et al. [45]. A positive bias of 2 $W/m^2$ has been found and the absolute bias equals 5 $W/m^2$ for the monthly mean SARAH-2 data.

### 2.1.2. Ground Stations

We used ground stations for the validation of the SESORA forecast. To ensure a high coverage of ground stations for our validation we wanted to use the pyranometers from the BSRN as well as the pyranometer set from the German Weather Service. The data from the BSRN are known for their high quality standards as the data is quality controlled twice, at the stations and at the World

Radiation Monitoring Center (WRMC) [46]. The high quality is reflected in the low standard deviation of 5 W/m$^2$ for the global irradiance. The BSRN stations used and further information are depicted in Table A1.

Additionally, we used a set of pyranometers from the German Weather Service. The data set consists of 34 stations and delivers global radiation at a temporal resolution of 1 min. These station are located in Germany only. Their standard deviation equals 3% of the respective total daily radiation. Corresponding information about these stations is listed in Table A2.

### 2.2. Optical Flow Method

In general, the optical flow describes the motion pattern between two sequential image frames of the same area. The result is a vector field where each vector is showing the movement of pixels from the first frame to the second [47].

We generally use the optical flow of Horn and Schunck [48] in a modification by Zach et al. [12,49]. The following constraint equation is the basis of all optical flow methods:

$$\nabla I \cdot \mathbf{u} + \frac{\partial}{\partial t} I = 0. \tag{1}$$

This equation describes a linear condition of the optical flow where $I(x(t), y(t), t))$ is the constant intensity between the two consecutive frames and $\mathbf{u}$ describes the two-dimensional velocity $(\dot{x}, \dot{y})$. A constant intensity is one of two major assumptions for the optical flow estimation. The second one is that neighboring pixels have to have similar motion [47]. The violation of these assumptions can lead to inconsistency and high, locally limited errors. An example for such a violation is the rapid formation of convective clouds between two consecutive frames. Thus, the first criterion, the intensity criterion, cannot be met because the reflectivity of the cloud top and its surrounding area is changing. The same effect can be seen when clouds or fog are dissolving as the value of CAL is changing in the opposite direction. An example case and further explanation about this topic can be found in Urbich et al. [12].

A scheme of the application of the optical flow is shown in Figure 1. The first step of the algorithm takes two subsequent satellite images of the effective cloud albedo as input data for the optical flow method. Here, we use the TV-$L^1$ method from the open source library OpenCV [17], as it is superior to the well known Farnebäck optical flow method [12,50]. The algorithm computes the estimated flow between the two frames, which is induced by the movement of clouds. The result is a vector field with a cloud motion vector for each pixel in the area. In the second step, the derived vectors are applied to the latter of the two observed CAL images to extrapolate the cloud movement into the future. In doing so, every pixel will be shifted, maintaining its original intensity. Thus their CAL value stays the same while their position can change. The step of applying the motion vectors to the satellite image can be repeated as often as required. In that case the vectors are applied to the latest forecast available in order to create a new one.

### 2.3. SPECMAGIC NOW

The SPECMAGIC NOW method is used in order to estimate the solar surface irradiance (SIS). In a first step the effective cloud albedo is being retrieved. It is derived from the geostationary satellite MSG by the reflectivity in the visible channel [30]. The visibility channel at 600 nm from SEVIRI on board of MSG is used for the calculation of CAL. The location of MSG is over the Equator at 0° latitude and longitude with a field of view from 80° S up to 80° N and from 80° E to 80° W. CAL is defined as the normalized difference between the all sky and clear sky reflectance in the 600 nm-visible channel of the satellite. The effective cloud albedo is equal to one minus the cloud transmission for values of CAL between 0 and 0.8 [12]. Above 0.8, this relation will be modified to consider the saturation and

absorption effects in optically thick clouds. The effective cloud albedo is derived from the normalized pixel reflectance, $\rho$, the clear sky reflectance, $\rho_{cs}$, and the maximal cloud reflectance, $\rho_{max}$ as follows:

$$\text{CAL} = \frac{\rho - \rho_{cs}}{\rho_{max} - \rho_{cs}}. \tag{2}$$

Here, $\rho$ is the observed reflectance for each pixel and time, and $\rho_{cs}$ is the clear sky reflectance, which is originally calculated according to an approach of Amilo et al. [51]. However, in this study, the original Heliosat approach is not used for the estimation of $\rho_{cs}$, instead it is derived from a database for spectral reflectance, as described in Müller et al. [39]. The maximal reflectance, max, is determined by the mean of the reflectance, $\rho$ above the 95th percentile and below the 99th percentile in the target region.

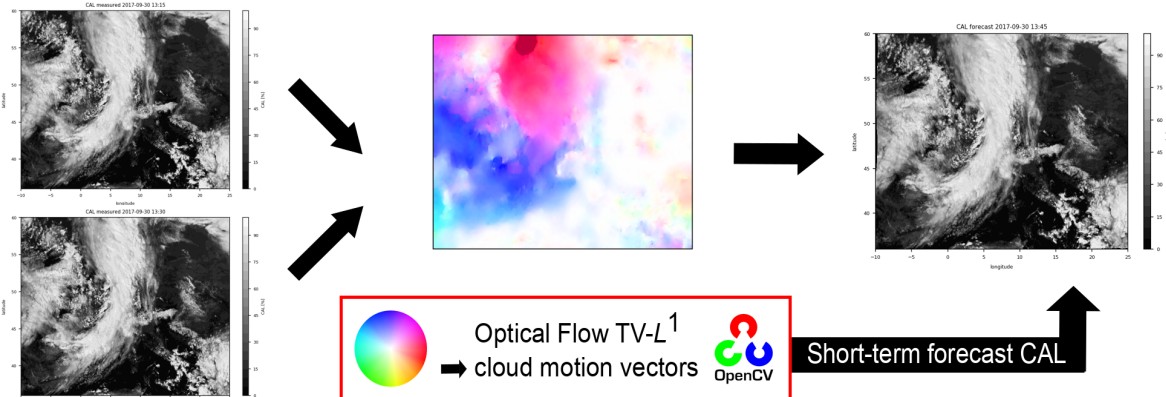

**Figure 1.** Scheme of the optical flow method TV-$L^1$ (Method based on total variation in the regularization term and the $L^1$-norm in the data fidelity term). Two subsequent images of the effective cloud albedo represent the input of the algorithm. The calculated motion vectors are then applied to the latter of the two images to extrapolate cloud albedo (CAL) into the future. For the sake of clarity the motion vectors are displayed with the help of the HSV (hue saturation value) color spectrum.

The effective cloud albedo is derived from satellite observations and is therefore a satellite-derived variable. This observable defines the cloud transmittance. For clouds with CAL in the range from 0 to 0.8, the following relation between CAL and the solar surface irradiance (SIS) is used:

$$\text{SIS} = (1 - \text{CAL}) \cdot \text{SIS}_{clear}. \tag{3}$$

Here, $\text{SIS}_{clear}$ is the clear sky irradiance at surface, which is calculated by a hybrid look-up table (LUT) approach. It is based on radiative transfer modeling and is described in detail in Müller et al. [39]. Equation (3) is used to estimate the satellite based solar surface irradiance for the observed CAL images as well as for the CAL nowcasting. For the CAL nowcasting the optical flow method described in Section 2.2 is applied to the observed CAL images. The optical flow method could also be applied to raw images followed by the estimation of CAL. However, this would lead to additional uncertainties induced by the surface reflectance. These effects are diminished by the application of the optical flow to the CAL images directly.

*2.4. Error Measures*

The solar surface irradiance nowcasting was verified in two different manners. The first part was done with SARAH-2 data, which enables a validation of every pixel of the whole given area. For this purpose we calculated the absolute difference between the forecast and the SARAH-2 data field for each pixel. On the basis of that we calculated three different error measures, which are used in the scope of energy meteorology as standard. The equations are as follows:

$$\text{bias} = \frac{1}{n} \sum_{i=1}^{n} \left( x_{\text{forecast}} - x_{\text{observation}} \right) \tag{4}$$

$$\text{absolute bias} = \frac{1}{n} \sum_{i=1}^{n} \left| x_{\text{forecast}} - x_{\text{observation}} \right| \tag{5}$$

$$\text{RMSE} = \sqrt{\frac{1}{n} \sum_{i=1}^{n} \left( x_{\text{forecast}} - x_{\text{observation}} \right)^2}. \tag{6}$$

Thus, we used the bias, absolute bias, and root mean square error for our validation. These errors were calculated for every examined case of this study. A list of all examined cases and their absolute bias can be found in Table A3. The table with the RMSE values can be found in the Appendix A (Table A4). Further, we plotted the absolute difference, which is depicted in Section 3.1. Moreover, on the basis of the previously mentioned error measures we calculated relative errors as follows:

$$\text{relative bias} = \frac{1}{n} \sum_{i=1}^{n} \left( \frac{x_{\text{forecast}} - x_{\text{observation}}}{\overline{x}_{\text{observation}}} \right) \tag{7}$$

$$\text{relative absolute bias} = \frac{1}{n} \sum_{i=1}^{n} \left| \frac{x_{\text{forecast}} - x_{\text{observation}}}{\overline{x}_{\text{observation}}} \right| \tag{8}$$

$$\text{relative RMSE} = \sqrt{\frac{1}{n} \sum_{i=1}^{n} \left( \frac{x_{\text{forecst}} - x_{\text{observation}}}{\overline{x}_{\text{observation}}} \right)^2}. \tag{9}$$

Here, $\overline{x}$ represents the mean of $x$.

The second part of the validation was performed with ground station data from BSRN and the German Weather Service (DWD). In that case we calculated the absolute difference between the observed radiation of the ground stations and the forecasted radiation of the nearest pixel to each corresponding station. Overall this results in 38 used pixels. The absolute bias and RMSE were calculated in the same way as above, however, on the basis of 38 single pixels. The results were plotted against the forecast time, together with the results of the nowcasting and the measurements of the stations (Section 3.2).

Based on the effective cloud albedo we determined a cloud mask where $CAL = 0.025$ marks the threshold between cloud and clear sky pixels. This value is similar to the usually used one in the literature, which equals 0.027 [52]. We found that for $CAL = 0.025$, the results were the most promising in reference to a significant distinction between cloud and clear sky pixels. Furthermore, the probability of detection was higher and the false alarm rate was lower with $CAL = 0.025$. This cloud mask was then used for verification with the SARAH-2 data to find the cause for high errors.

On the basis of this cloud mask we calculated the elements of the contingency table for the forecast and the observation, which are hit, missed, false alarm, and correct negative (Table 1). These elements were furthermore displayed in a map, which can be seen in Section 3.1. The elements were also used to calculate the probability of detection (POD) (Equation (10)) and the false alarm rate (FAR) (Equation (11)) as follows:

$$\text{POD} = \frac{a}{a+c} \tag{10}$$

$$\text{FAR} = \frac{b}{a+b}. \tag{11}$$

**Table 1.** Contingency table with the elemtents hit, missed, false alarm (fa) and correct negative (cn).

|  |  | Observation | |
| --- | --- | --- | --- |
|  |  | **Cloud** | **No Cloud** |
| Forecast | Cloud | $a$ = hit | $b$ = fa |
|  | No Cloud | $c$ = miss | $d$ = cn |

## 3. Results

Seventeen different cases were examined in this study based on different weather situations for the months of August until October, in 2017. The same cases have been already examined in Urbich et al. [12] concerning the effective cloud albedo. A list of all cases and their error measures can be found in Tables A3 and A4. More cases in different seasons or years may not deliver additional information, as clouds play a dominant role in solar radiation forecasts. Therefore we assume that the diversity of weather situations in this study should cover all relevant cloud types for a solar radiation forecast. In the following section two particular cases out of 17 in total will be discussed for the sake of clarity. The following cases show different weather situations and should be seen as representatives for the remaining cases.

### 3.1. SARAH-2

In Figure 2 the solar surface radiation is depicted for two different cases. The first case (Figure 2a,b) is 29 August 2017. The general weather situation was a high pressure system over central Europe. The second case (Figure 2c,d) is 30 September 2017 and in that case there was a low pressure system over western Europe and a front was passing Germany during the day. These cases were selected due to their different occurrence of clouds and solar radiation. A 255 min (4 h 15 min) forecast is shown in Figure 2b,d with the corresponding estimated SARAH-2 data set in Figure 2a,c.

The forecasted radiation for the first case shows promising results compared to the SARAH-2 data. All in all, general structures are well met, as well as the height of the values themselves. The cloud structure over the North Sea is also shown by the nowcasting, however with less detail and a light displacement. This nowcasting consists of the optical flow of effective cloud albedo and the calculation of the radiation with SPECMAGIC NOW. Therefore, errors can be caused by two separate sources. That the cloud structure shows less details is probably caused by the effective cloud albedo nowcasting. Further, the broken clouds over Spain are displaced in the nowcasting. In particular, smaller clouds are more affected by the algorithm, as the fraction of cloud edges in relation to the inner part of the cloud is larger. Cloud borders can cause errors due to wrong advection and cloud dissipation or formation. Since the nowcasting works without any kind of boundary conditions or data beyond the depicted area there will always be some part of the plot with no data. This part is displayed in black. It grows with increasing forecast time because the edge is moving inwards. However, this is not a problem for the application of the SESORA forecast since the distribution (DSO) and transmission system operators (TSO) who will use the forecast only need the area of Germany and the surrounding regions.

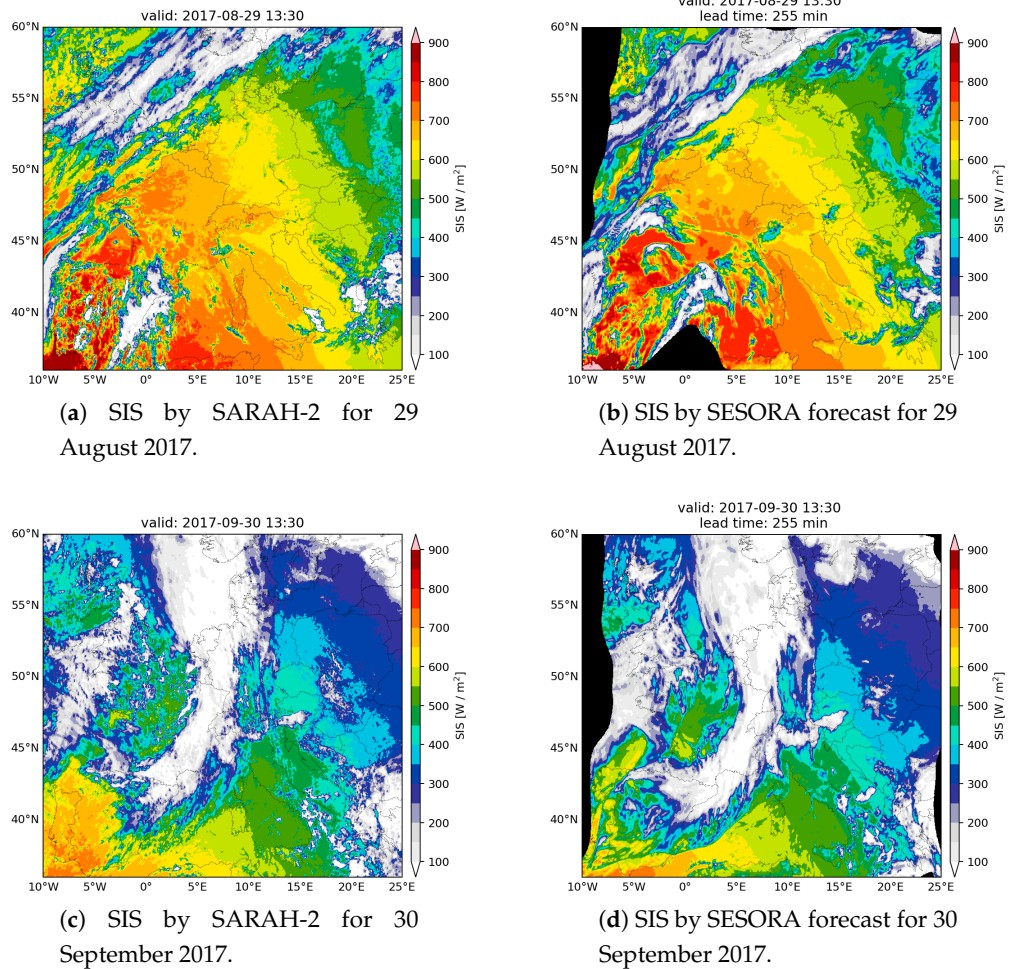

**(a)** SIS by SARAH-2 for 29 August 2017.

**(b)** SIS by SESORA forecast for 29 August 2017.

**(c)** SIS by SARAH-2 for 30 September 2017.

**(d)** SIS by SESORA forecast for 30 September 2017.

**Figure 2.** Plot of solar surface irradiation for 29 August 2017 (upper) and 30 September 2017 (lower). (**a**,**c**) show the estimated SIS (Surface Incoming Shortwave Radiation) by SARAH-2 (Surface Solar Radiation Data Set - Heliosat) and (**b**,**d**) show the forecasted SIS by SESORA (Seamless Solar Radiation) after 255 min of forecast time.

Similar results can be observed in the second case. Except for smaller details, the position of clouds and the height of the radiation values are comparable. The structure of the front in the SARAH-2 data consists of more small clouds, which may have blurred out due to the long forecast time and the southern end of the front advecting too slowly in the nowcasting. Moreover, there is a cloud hole over southern Germany with higher radiation values than in the nowcasting, which is a result of an optically too thick cloud calculated by SPECMAGIC NOW. In general, one can say that cloud borders pose the biggest problem to the radiation forecast, as has been discussed before. Thus, the more small clouds, the higher the incidence of problematic edge regions, and the higher the errors will be.

To prove and visualize the previously seen differences, the absolute bias was calculated according to Equation (5), between the solar surface radiation nowcasting and the SARAH-2 data set. The results are displayed in Figure 3. The regions with higher errors correspond to the above mentioned regions. The cause of these errors are missing cloud structures, for instance over Austria, as well as incorrectly forecasted cloud edges, as can be seen over the North Sea and Spain (Figure 3a). These errors grow as usual with increasing forecast time. The absolute bias for 255 min equals 92 W/m$^2$ and the RMSE equals 143 W/m$^2$. As this is a nowcasting of solar surface irradiance, the values, and also the errors, decrease when the sun sets. This effect cannot be seen at this stage of the forecast, however it can be observed in Figure 4.

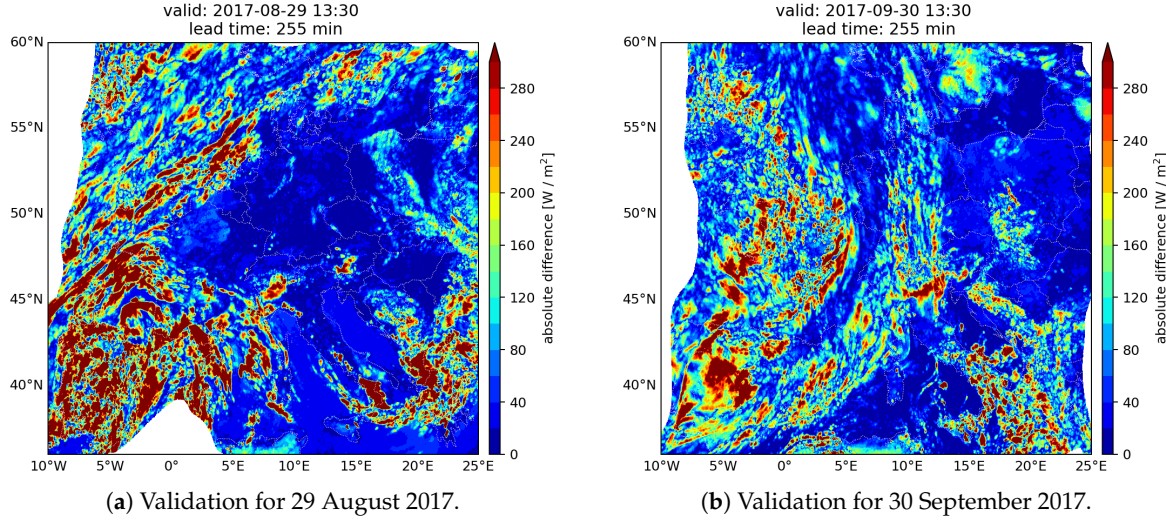

(**a**) Validation for 29 August 2017.        (**b**) Validation for 30 September 2017.

**Figure 3.** Validation of the solar surface irradiation nowcasting with SARAH-2 data for 29 August 2017 (**a**) and 30 September 2017 (**b**). Depicted is the absolute difference between the SIS nowcasting and the SARAH-2 data set for a 255 min forecast.

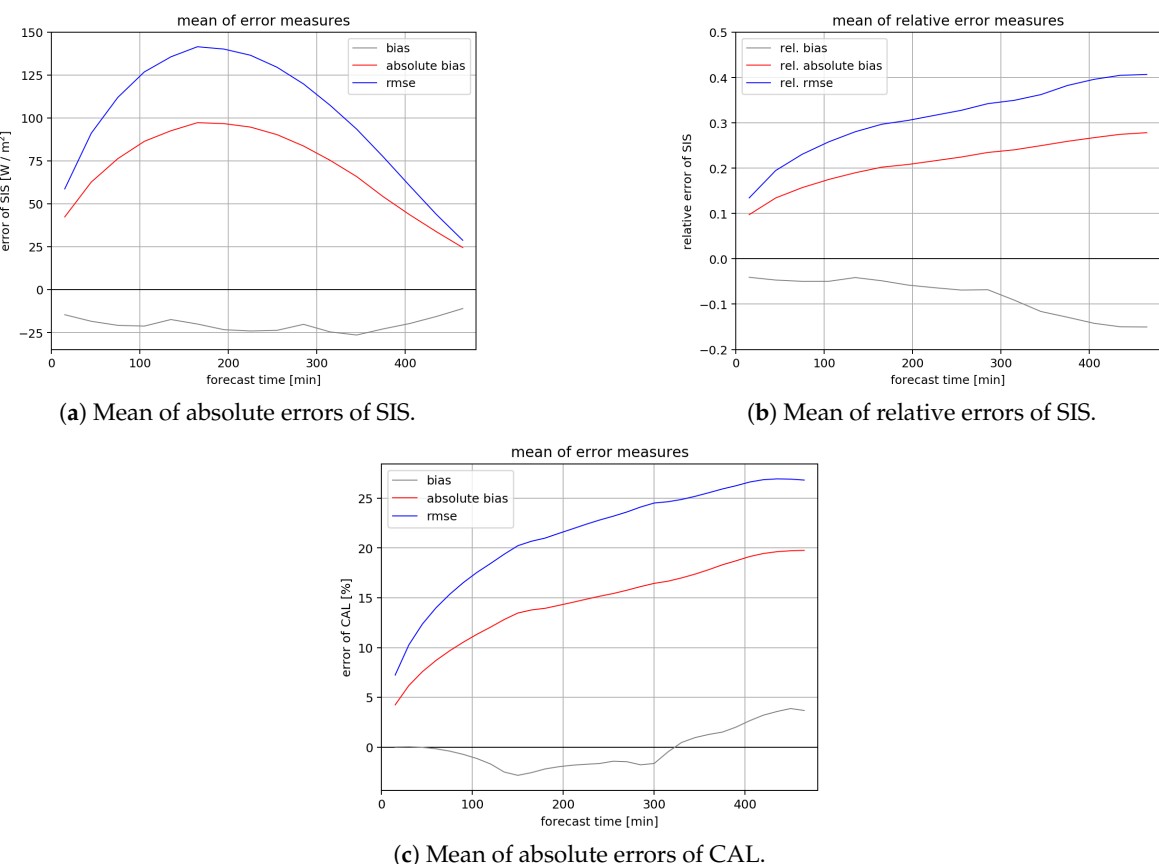

(**a**) Mean of absolute errors of SIS.        (**b**) Mean of relative errors of SIS.

(**c**) Mean of absolute errors of CAL.

**Figure 4.** Plots of the mean error measures of all cases against forecast time for absolute (**a**) and relative errors (**b**) of SIS as well as the absolute errors of CAL (**c**). The validation of the solar surface irradiance was performed with SARAH-2 data by the CM SAF (Climate Monitoring Satellite Application Facility) and the validation of the effective cloud albedo was done with the effective cloud albedo itself.

In the case of 30 September 2017 the validation appears different (Figure 3b). One of the issues in the nowcasting was the broken prefrontal clouds. Due to a generally less detailed effective cloud albedo, nowcasting the structure of these clouds looks different. This led to a slightly incorrect nowcasting of solar radiation between the clouds. Another problem is the back of the front. Smaller cloud structures are missing as well. What can be observed in Figure 3b are many smaller regions of errors over Germany, which are not as big as the error behind the front over France. An absolute bias of 79 W/m$^2$ and a RMSE of 112 W/m$^2$ have been found for this case.

In Figure 4 the mean of all error measures for all cases is plotted against forecast time. In Figure 4a there are the absolute measures and in Figure 4b there are the relative error measures. A list of all cases examined can be found in Tables A3 and A4. All forecasts were initiated at 09:15 UTC and the maximum forecast time was 480 min. Depicted are the bias (gray), the absolute bias (red), and the root mean square error (blue), respectively. What can be observed in Figure 4a is that the absolute bias and RMSE grow with increasing forecast time until approximately 180 min. After that, both error measures decrease again due to sunset. The behavior of the bias looks different because it does not represent an absolute error but rather a tendency. Therefore, one can say that for all times the nowcasting underrates solar irradiation, thus the estimated solar radiation by SARAH-2 delivers higher SIS values (Equation (4)). This is a result of SPECMAGIC NOW, which currently calculates the optical thickness of clouds higher than it should, due to $\rho_{cs}$ being too low (Equation (2)). In fact, this kind of error can be fixed quite quickly, and an update of SPECMAGIC NOW is already planned, where $\rho_{cs}$ will be adapted to reduce the bias. The relative errors show, as expected, a different behavior. As the errors are normed by the mean of the observed SIS values, the sunset does not play a role in this case (Equations (7)–(9)). The relative absolute bias and the relative RMSE rise with increasing forecast time. The slope of these two curves decreases with increasing forecast time, which results in a slower growth of the relative errors. The maximum of the RMSE is 0.41 and the maximum value of the relative absolute bias is 0.28. For the sake of a forecast validation without the influence of the solar altitude the mean of all error measures of the effective cloud albedo is depicted in Figure 4c. The absolute bias and RMSE show the same behavior as the absolute bias and RMSE of the relative errors of SIS. The bias is negative for the first 300 min and turns positive afterwards.

Another method of verifying the quality of the SESORA forecast is a linear regression for all examined cases. Therefore the forecasted values of solar radiation were plotted against the observed radiation with the help of the SARAH-2 data set for each pixel in every frame and for each case dependent upon the forecast time. The results are shown in Figure 5. Moreover, a standardized regression was done where the solar zenith angle of the forecasted and observed solar radiation was corrected. Thus, sunset is less important for the quality of the forecast. As expected the distribution in Figure 5a–c gets broader with increasing forecast time and the values of SIS get smaller in the observation as well as in the nowcasting because of the sunset. Most of the data points are lying on the diagonal whereby the distribution is split into a maximum for smaller and a maximum for higher values of solar irradiance. This behavior remains unchanged throughout the forecast. The slope is smaller than 1 for all forecast times, which underlines the negative bias found in Figure 4. As can be seen in Figure 5 for the forecast times 135 min and 255 min, the observed values are higher than the forecasted SIS values especially for small values. As a comparison, the bias for all forecast times until 400 min was $\approx -25$ W/m$^2$. Looking at the spread we can see that there are more small values of solar radiation, and therefore the linear regression does not begin at the origin as it is shifted upwards. That is also the reason for general slope values below 1 for all forecast times. The quality of the linear regression is represented by the R$^2$-value, which is displayed in the lower right corner of each linear regression plot. After 15 min the R$^2$ remains quite high with a value of 0.94. After 135 min we found a R$^2$ of 0.72. A forecast time of 2 h is a typical length for nowcasting, thus it is a common forecast time for comparisons with other publications. Sirch et al. found a R$^2$-value of 0.71 for a DNI (Direct Normal Irradiance) nowcasting after 120 min in March and a value of 0.64 in July [29]. It can be observed that the forecast quality improves when the angle of the sun is being corrected. This underlines the fact

that the bias of $\approx -25 \text{ W}/\text{m}^2$ found in the validation with the SARAH-2 data arises from a systematic error in SPECMAGIC NOW. The p-value for all forecast times was smaller than $1 \cdot 10^{-300}$, which shows the high significance of the distribution. It also proves that the distribution of the data is non-normal thus we can reject the null hypothesis.

It is essential to distinguish between different error sources in a nowcasting, for the improvement of the algorithm, however, the more steps of computation are involved, the more complications may be found. For the SESORA forecast we found a systematic error in the calculation of the solar surface irradiance, which can be clearly seen in the constant bias in Figure 4. This bias can be corrected by adjustments in SPECMAGIC NOW and it is not related to TV-$L^1$. For the remaining part of our algorithm, which is the nowcasting of the effective cloud albedo, we divided the errors into cloudy pixels and clear sky pixels (Section 2.4). The idea is to detect errors resulting from convection or advection separately. The results are shown in Figure 6.

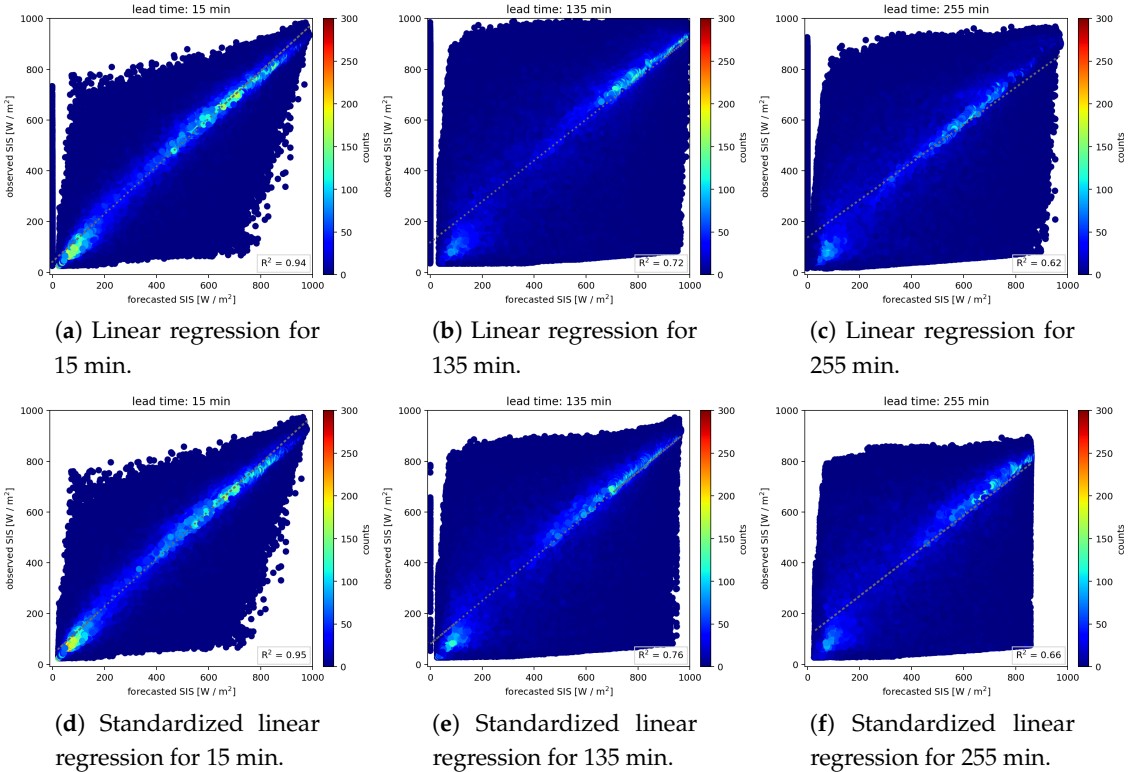

(**a**) Linear regression for 15 min.

(**b**) Linear regression for 135 min.

(**c**) Linear regression for 255 min.

(**d**) Standardized linear regression for 15 min.

(**e**) Standardized linear regression for 135 min.

(**f**) Standardized linear regression for 255 min.

**Figure 5.** Linear regression of the forecasted and observed absolute solar surface irradiation and standardized solar surface irradiation. Depicted are the results for 15 min (**a**,**d**), 135 min (**b**,**e**), and 255 min (**c**,**f**) of forecast time respectively. $R^2$ is printed in the lower right corner of every figure.

In Figure 6c,f, the errors that are marked miss and false alarm (fa) mostly arise from wrong advection of the optical flow algorithm. When the TV-$L^1$ method calculates a cloud motion too slowly or too quickly, this leads to errors at the edge of the clear sky area. In the cloudy area this error can occur as well, however we cannot find them with our analysis. If our algorithm calculates the cloud motion too slowly we will get a miss and if the motion is calculated too quickly we will get a false alarm. However, in general we detect more misses than false alarms. Moreover, the errors rise with increasing forecast time as can be seen in Figure 6f. The magnitude of errors cannot be extracted from this graphic, although when we take Figure 6b into account we can see that the errors due to wrong advection are rather small. The errors in Figure 6b,e are small in general. Thus, the errors with the highest magnitude are caused by clouds. These kind of errors can be detected in Figure 6a,d and they all are caused by a change of intensity of the effective cloud albedo over time. As was already

mentioned in Urbich et al. [12], the change of the pixel intensity over time is a major issue for the optical flow. These errors have the highest magnitude and appear more frequently than errors due to wrong advection. As usual, all errors grow with increasing forecast time (Figure 6d–f).

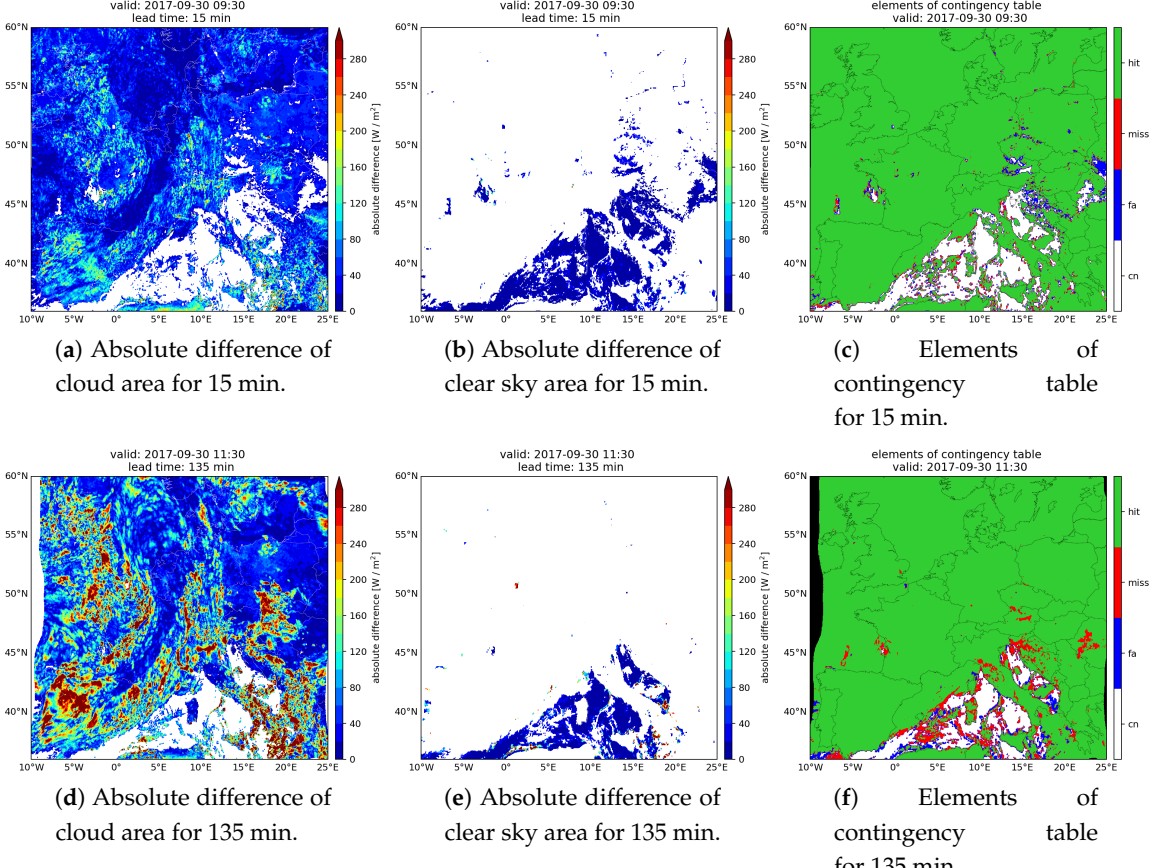

(**a**) Absolute difference of cloud area for 15 min.

(**b**) Absolute difference of clear sky area for 15 min.

(**c**) Elements of contingency table for 15 min.

(**d**) Absolute difference of cloud area for 135 min.

(**e**) Absolute difference of clear sky area for 135 min.

(**f**) Elements of contingency table for 135 min.

**Figure 6.** Results of the validation for 30 September 2017 with SARAH-2 and an additional cloud mask. Depicted are the absolute difference between the nowcasting and the SARAH-2 data (**a**,**b**,**d**,**e**), as well as a map of the elements of the contingency table (**c**,**f**). These results are shown for 15 (upper) and 135 min (lower), respectively.

### 3.2. Ground Stations

In Figure 7a the nowcasting for 255 min of the solar surface irradiance is displayed for 29 August 2017. Overall, the measurements of the ground stations show agreement with the nowcasting in this case. For this type of validation we must keep in mind that the geometry of these two measurements is completely different. MSG is located at 0° longitude and latitude, and thus its viewing angle to the surface in the area of Europe is slant. In contrast, pyranometers are standing at the surface and only measure the radiation above them. Furthermore, we are comparing point observations with area integrals of approximately 16 km$^2$ (in the area of Germany). This is especially difficult if there are sub-pixel scattered clouds. These effects add uncertainties that are not caused by shortcomings of the nowcasting method. So, in some cases the value of the ground station does not seem to fit to the forecasted radiation but this could be either an artifact of the geometry of the satellite or the comparison of point observations with areas.

Figure 7b shows the same content as in Figure 7a for 30 September 2017. Again, the inward moving edge on the left side of the figure can be observed. The agreement of the stations and the nowcasting is not as good as in the Figure 7a of 29 August 2017. The station of Palaiseau (marked by a red circle) shows higher values than the nowcasting. Even in the surrounding area such high

values between 600 and 650 W/m$^2$ cannot be found. The same can be observed for the station of Arkona (marked by a red circle), which also measured values between 400 and 450 W/m$^2$ however the nowcasting shows values below 350 W/m$^2$.

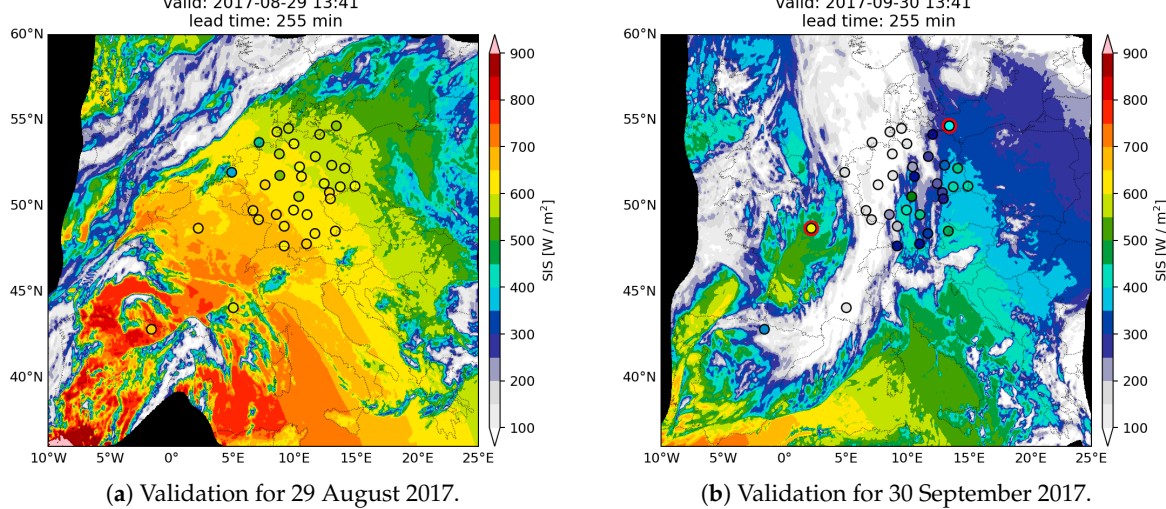

**Figure 7.** Validation of the solar surface irradiance nowcasting with ground stations by BSRN (Baseline Surface Radiation Network) and DWD (Deutscher Wetterdienst) for 29 August 2017 (**a**) and 30 September 2017 (**b**). Depicted is the nowcasting of SIS as contour plot and the measured value of the ground stations is given in the black circles. The colorbar matches both figures. The forecast time equals 255 min. The red circles in (**b**) mark the stations Arkona and Palaiseau.

The error measures of the validation with ground stations are depicted in Figure 8. Displayed is the mean of all 17 cases against forecast time for the area of Europe. The corresponding solar surface irradiance value of the nowcasting (red), as well as the one of the ground stations (blue) is plotted against forecast time for overall 480 min. We only selected the satellite pixels of the nowcasting that corresponded to a pixel of a ground station. Although it is a common approach to take the mean of a 3 × 3 pixel area around the pixel of the ground station, we decided to take only one pixel to achieve a realistic error measure for the purpose of PV systems. This nowcasting aims to warn PV system operators of grid instability and a realistic measure of the uncertainty of our forecast is essential. With the absolute difference of the nowcasting and the observation the root mean square error (black) and absolute bias (gray) were calculated (Figure 8a). We also calculated the respective relative errors by normalizing the absolute errors with the mean of the observed solar radiation at the surface that was measured by the pyranometers (Figure 8b).

The solar radiation of the nowcasting shows smaller values than the ground stations until approximately 250 min. Nevertheless, both the nowcasting and the observation show a similar behavior and a decrease of solar irradiance with increasing forecast time. The decrease of SIS can be observed due to the sunset and due to the fact that we work with products from the visible channel. The curves do not significantly differ from each other, which also results in small errors for the whole nowcasting. Furthermore the RMSE does not exceed 200 W/m$^2$. A slight maximum can be observed between 100 min and 200 min forecast time. The behavior of the error curves differs slightly from the RMSE and absolute bias in Figure 4 where the maximum is more distinct. Furthermore, the curve in Figure 4 shows less fluctuations but the height of the errors is on a comparable level. Nevertheless, the visual validation that can be seen in Figure 7 shows that the solar irradiation nowcasting matches most of the pyranometers. The relative errors in Figure 8b show the same behavior as the relative errors calculated for the validation with the SARAH-2 data in Figure 4. Until approximately 400 min, both the relative absolute bias and the relative RMSE increase with increasing forecast time. The relative RMSE

reaches higher values after 400 min of forecast time because the majority of the stations measured $0 \, W/m^2$, and certain stations did not measure any data at all. As a consequence, the stations that did measure solar radiation at the surface have a higher impact on the result. This led to a higher difference between the nowcasting and the observation, which, after Equation (9), results in a higher relative RMSE, or even in values above 1. in addition to the rising errors after 400 min of forecast time, the height of the relative errors is in the same order as the relative errors from the SARAH-2 validation, which are displayed in Figure 4.

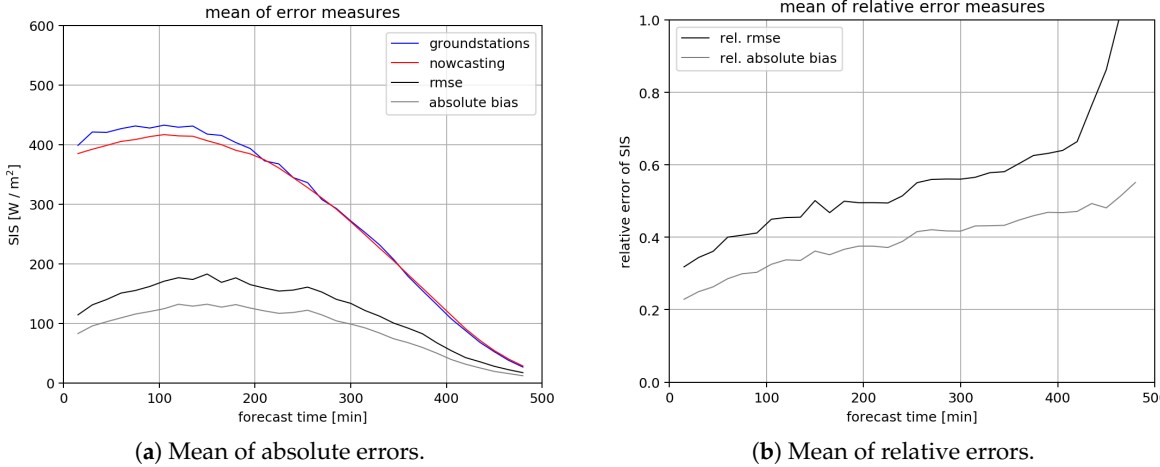

(**a**) Mean of absolute errors. (**b**) Mean of relative errors.

**Figure 8.** Plots of the mean error measures of all cases against forecast time for absolute (**a**) and relative errors (**b**). The validation of the solar surface irradiance was performed with ground stations by BSRN and DWD.

## 4. Discussion

In this work we presented many different validation methods to prove the quality of the SESORA forecast and to find out more about the errors that occur in our nowcasting.

The use of the visible channel of MSG leads to the issue that we cannot calculate motion vectors during the night. Since the SESORA forecast calculates the solar surface irradiance we are not interested in the night itself, however, it would be useful to be able to give a forecast for the early morning hours when the sun rises. This problem may be solved by a combination of the visible with the infrared channel. An advantage of the solar zenith angle dependency of SIS is that the SESORA forecast improves quantitatively when the sun sets because as the values of SIS decrease the errors of the nowcasting decrease as well. Due to this fact, NWP does not necessarily deliver better results after 4 h, when one looks at the results in the evening hours. Thus, the point of interception gets shifted back to longer forecast times. For 13 of the 17 cases discussed in this study, a comparison of the SESORA forecast with different NWP models and persistence has been performed within a master's thesis [53]. In this thesis it is shown that the point of intersection between our solar surface irradiance nowcasting and the IFS model forecast by the ECMWF (European Centre for Medium-Range Weather Forecasts) is 2:45 h with a deviation of 17 min for the RMSE [36]. However, the intersection point with the NWP depends largely on the used model. For the ICON model by the German weather service the intersection point is 04:32 h with a standard deviation of 58 min [53]. The nowcasting performs significantly better than persistence for all forecast scales, the margin rising with increasing forecast time. Figure 9 contains the results of the RMSE of the different forecasts averaged over 13 of the investigated test cases in this study. The relative errors of SIS increase with increasing forecast time, which is an expected behavior of all forecasts. This is of course important to evaluate the quality of the forecast however for the end users an absolute error is more useful. The users of the SESORA forecast are transmission and distribution system operators as well as direct marketers, and they may use it for

trading and grid security, and in this case absolute errors are useful as they are more direct. All in all, both errors have advantages and disadvantages for each specific application.

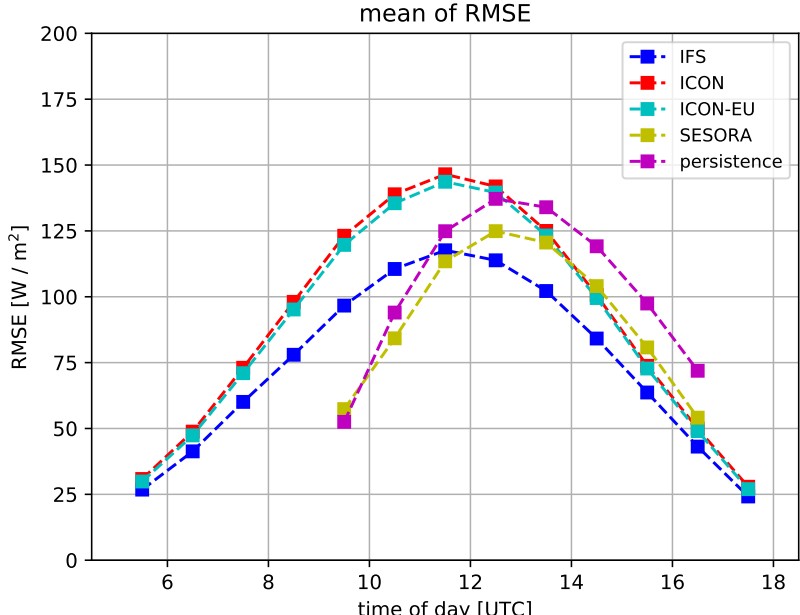

**Figure 9.** Plot of the mean of RMSE of SIS for 13 cases against the time of day. The comparison is shown for IFS (Integrated Forecasting System) by the ECMWF (European Centre for Medium-Range Weather Forecasts), ICON (ICOsahedral Nonhydrostatic) and ICON-EU (ICON only for Europe) by the German Weather Service (DWD), SESORA and persistence. The validation was performed with SARAH-2 data by the CM SAF. For SESORA and persistence the forecast was initiated at 09:15 UTC. They have a maximum lead time of 8 h. IFS is available every 12 h, ICON/ICON-EU every 3 h. It was taken from the model run, which would be available in reality, thus initiation time plus the time for the model run to finish (IFS takes 6 h; ICON/ICON-EU takes 3 h). Image adapted from [53].

In the literature there are many possibilities of validating a solar irradiance nowcasting. Many approaches for such a nowcasting with the use of cloud motion vectors have been proposed [25–29]. A comparison with other studies is not always simple, as the region or the validation method can differ. Schroedter-Homscheidt and Gesell for example proposed a nowcasting for DNI over Spain and validated it by calculating the bias. After approximately 5 h the bias equaled 130 W/m$^2$ [27]. Nonnenmacher and Coimbra as well as Gallucci et al. used the RMSE as their validation method for the region of San Diego and Italy. After 2 h Nonnenmacher and Coimbra got a RMSE of approximately 145 W/m$^2$ and Gallucci et al. mention a RMSE of 147 W/m$^2$ [26,28]. For the SESORA forecast we found a RMSE of 136 W/m$^2$ after 135 min over the area of Europe. In this comparison the area and the season play a major role so the three nowcastings are probably of similar quality. Another way of validating a nowcasting can be the probability of detection for a cloud mask. In our case we only distinguished between cloud and clear sky however Sirch et al. proposed a differentiation of upper and lower clouds [29]. They found a POD of 85–90% in dependency of the cloud type after 2 h over the area of Spain. The POD of the SESORA forecast after 135 min equals 84%, though our cloud mask is simply based on a effective cloud albedo threshold. Further, Sirch et al. performed a linear regression for DNI for the months March and July. R$^2$ equals 0.71 and 0.64 after 2 h while for our study R$^2$ is 0.72 after 135 min for SIS. Again, these comparisons are difficult and can only serve as a point of reference. All of the above discussed results can be found in Table 2.

**Table 2.** List of comparable results (Direct Normal Irradiance (DNI), Global Horizontal Irradiance (GHI), Surface Incoming Shortwave Radiation (SIS)) with different error measures (Root Mean Square Error (RMSE), Probability of Detection (POD)) from other publications and this study.

| Author | Variable | Area | Measure | Lead Time (min) | Value |
|---|---|---|---|---|---|
| Schroedter-Homscheidt and Gesell [27] | DNI | Spain | Bias | 300 | 130 W/m$^2$ |
| Nonnenmacher and Coimbra [26] | GHI | San Diego | RMSE | 120 | 145 W/m$^2$ |
| Gallucci et al. [28] | SIS | Italy | RMSE | 120 | 147 W/m$^2$ |
| this study | SIS | Europe | RMSE | 135 | 136 W/m$^2$ |
| Sirch et al. [29] | DNI | Spain | POD | 120 | 85–90% |
| this study | SIS | Europe | POD | 135 | 84% |
| Sirch et al. [29] | DNI | Spain | $R^2$ | 120 | 0.71 |
| this study | SIS | Europe | $R^2$ | 135 | 0.72 |

## 5. Conclusions and Outlook

In this work we presented a validation of a short-term forecast, also called nowcasting, based on the optical flow of effective cloud albedo for solar surface irradiance. The basis of our nowcasting works with the optical flow method TV-$L^1$ by OpenCV. The effective cloud albedo that can be retrieved by the reflectivity of the visible channel of MSG serves as input for our algorithm. As a result we have a field of cloud motion vectors that describe the apparent motion between two consecutive satellite images. These vectors are applied to the latter of the satellite images to create the first forecast step. The stage of applying the motion vectors can be repeated as often as desired to generate another forecast step. In that case the vectors are being applied to the latest available forecast step. Finally, by means of SPECMAGIC NOW the solar surface irradiance is calculated for every CAL forecast step to create the final SIS nowcasting.

We performed the validation with SARAH-2 data as well as with ground stations from the BSRN and the DWD. In both cases we calculated three different error measures, namely the bias, absolute bias, and root mean square error (Section 2.4) These error measures were calculated as absolute and relative errors respectively. All nowcastings shown in this study had a maximum forecast time of 480 min, which conflicts with the sunset. As a consequence the values of the solar surface irradiance, their absolute errors decreased with increasing forecast time. The maximum of the absolute bias and the RMSE can be found approximately after 180 min of forecast time if one considers Figure 4. If we look at the calculated values, which are available every 30 min due to the validation with the SARAH-2 data set we find the highest errors for 165 min. The absolute bias for 165 min equals 97 W/m$^2$ and the RMSE equals 142 W/m$^2$ for the same time. Relative errors are shown to grow with increasing forecast time and the maximum of the relative RMSE equals 0.41 after 480 min. A large issue with relative errors is high background radiances. As a consequence, resulting errors could be higher than large relative errors of low background radiances. Thus, even a small relative error could have an impact on the grid stability in that case where the forecast error affects a large number of photovoltaic systems. However, the SESORA forecast works best for high pressure system situations when there are only a few clouds and the solar irradiance is high (Figure 3). Therefore, the risk of providing a bad forecast for the TSOs and DSOs is rather small. Moreover, the biggest errors are caused by convection but convective cells are rather local phenomena and it is very unlikely that enough photovoltaic systems are simultaneously affected by this type of error in Germany, that they would induce grid instability. Furthermore, after the SPECMAGIC NOW update the constant bias of solar irradiance should be significantly lower due to corrections in the clear sky calculation. For the upcoming merge with the NWP, a quality check is planned to determine the intersection point with the nowcasting. Thus, if the errors of the nowcasting will actually rise too quickly the merge with the NWP will be performed earlier to prevent large forecast errors and as a consequence of grid instability. Absolute errors of the

effective cloud albedo rise as well as the relative errors of solar irradiance with increasing forecast time. The results for the validation with ground stations showed similar results with slightly higher errors (relative RMSE $\approx 0.6$ after 480 min). The linear regression proved the bias of $\approx -25$ W/m$^2$ found in the validation with the SARAH-2 data. There is a planned update of SPECMAGIC NOW that should reduce this bias by adapting the optical thickness of clouds. By reducing the systematic error of the SESORA forecast the overall results will improve. Errors that occur due to the optical flow method TV-$L^1$ cannot be corrected as easily. The effect of growing errors with increasing forecast time is represented by the spread of the data points in the regression. Furthermore, the effect of the sunset can be corrected by the cosine of the solar zenith angle. This correction also improved the quality of the forecast as can be seen in the increased R$^2$ value. Overall, the results appear promising and the errors of the examined cases are small. The largest errors occur at cloud edges or in the case of formation or dissipation of clouds. Since the algorithm of our nowcasting is built on extrapolation it neglects convective or dissipative processes.

Future work can implement features that allow curved trajectories for cloud pixels. Moreover these features could allow the formation and dissipation of clouds by taking into account changing intensities in the optical flow estimation. For both additions it is necessary that the optical flow method uses more than two satellite images. Therefore, we will adjust the algorithm such that multiple frames will serve as import data in the optical flow method to allow a longer history of cloud movements. Another aim of ours is the use of the NWP for later hours of our nowcasting where a merge between these two is planned in the range of an intersection point. The transition of the higher forecast quality from the nowcasting to NWP depends on the weather situation, initiation time and maybe other parameters. However, in our opinion the point of intersection between the sinking quality of the nowcasting and the increasing quality of the NWP would probably lie between 3 to 4 h. After that, the loss of details due to the optical flow method becomes too high and the NWP might deliver better results. With a merge between the nowcasting and the NWP we would develop a seamless product that always uses the best available forecast at each time step for a 12 h solar surface irradiance forecast. Some leading experts like Lorenz and Wolff have already shown that seamless products deliver even better results than the NWP after the point of intersection as the combination leads to an improvement of the forecast quality [8,9]. Thus, the final product would not only be seamless but it would also show a higher quality than single forecast products.

The software of SPECMAGIC NOW, as well as the optical flow method TV-$L^1$, are both open source. This validation study works as an indication for future works of other scientists to use one or both parts for their own research. This way, PV forecasts can simply be constructed out of smaller software blocks and adjusted to the needs of the operator.

**Author Contributions:** I.U. performed the validation of this work. R.M. developed the approach for the retrieval of the effective cloud albedo and the clear-sky model SPECMAGIC NOW. J.B. participated as the mentor of the work. All authors contributed to the writing of the manuscript.

**Funding:** This reasearch was funded by Gridcast, a project by the Federal Ministry for Economic Affairs and Energy (Bundesministerium für Wirtschaft und Energie, BMWi).

**Acknowledgments:** We thank Jörg Trentmann and Uwe Pfeifroth for providing the SARAH-2 and BSRN data for our validation. Thanks to Michael Mott, Manuel Werner and Nils Rathmann for the introduction and support regarding the POLARA framework, which was developed by the department of radar meteorology at the German Weather Service.

**Conflicts of Interest:** The authors declare no conflict of interest.

## Appendix A

**Table A1.** List of BSRN pyranometer stations in the region of Europe. The stations that did not measure the radiation or deliver the data for the period of our study were not listed.

| Event | Label | Location | Latitude (°) | Longitude (°) | Elevation (m) |
|---|---|---|---|---|---|
| Cabauw | CAB | Netherlands | 51.9711 | 4.9267 | 0.0 |
| Carpentras | CAR | France | 44.0830 | 5.0590 | 100.0 |
| Cener | CNR | Spain | 42.8160 | −1.6010 | 471.0 |
| Palaiseau | PAL | France | 48.7130 | 2.2080 | 156.0 |

**Table A2.** List of DWD pyranometer stations.

| Location | DWD-ID | Latitude (°) | Longitude (°) | Elevation (m) |
|---|---|---|---|---|
| Arkona | 183 | 54.6791 | 13.4342 | 42.0 |
| Braunlage | 656 | 51.7233 | 10.6021 | 607.3 |
| Braunschweig | 662 | 52.2914 | 10.4464 | 81.4 |
| Bremen | 691 | 53.0445 | 8.7985 | 4.3 |
| Chemnitz | 853 | 50.7912 | 12.8719 | 418.0 |
| Dresden | 1048 | 51.1279 | 13.7543 | 227.0 |
| Fichtelberg | 1358 | 50.4283 | 12.9535 | 1213.0 |
| Geisenheim | 1580 | 49.9859 | 7.9548 | 110.2 |
| Görlitz | 1684 | 51.1621 | 14.9505 | 238.0 |
| Hamburg | 1975 | 53.6331 | 9.9880 | 14.1 |
| Hohenpeissenberg | 2290 | 47.8009 | 11.0108 | 977.0 |
| Konstanz | 2712 | 47.6774 | 9.1900 | 442.5 |
| Leipzig | 2928 | 51.3150 | 12.4462 | 138.0 |
| Lindenberg | 3015 | 52.2084 | 14.1179 | 98.0 |
| Bad-Lippspringe | 3028 | 51.7854 | 8.8387 | 157.0 |
| Lüdenscheid | 3098 | 51.2451 | 7.6424 | 386.7 |
| Meiningen | 3231 | 50.5611 | 10.3771 | 450.0 |
| Norderney | 3631 | 53.7123 | 7.1519 | 11.5 |
| Nuremberg | 3668 | 49.5030 | 11.0549 | 314.0 |
| Potsdam | 3987 | 52.3812 | 13.0622 | 81.0 |
| Rostock | 4271 | 54.1801 | 12.0805 | 4.0 |
| Saarbrücken | 4336 | 49.2128 | 7.1077 | 320.0 |
| Sankt-Peter-Ording | 4393 | 54.3279 | 8.6029 | 4.9 |
| Schleswig | 4466 | 54.5275 | 9.5486 | 42.7 |
| Seehausen | 4642 | 52.8911 | 11.7296 | 21.0 |
| Stuttgart | 4928 | 48.8281 | 9.2000 | 314.3 |
| Trier | 5100 | 49.7478 | 6.6582 | 265.0 |
| Weihenstephan | 5404 | 48.4024 | 11.6945 | 477.1 |
| Weissenburg | 5440 | 49.0113 | 10.9319 | 439.3 |
| Würzburg | 5705 | 49.7702 | 9.9577 | 268.0 |
| Zugspitze | 5792 | 47.4208 | 10.9847 | 2964.0 |
| Fürstenzell | 5856 | 48.5451 | 13.3530 | 476.4 |
| Mannheim | 5906 | 49.5090 | 8.5540 | 98.0 |
| Schneefernerhaus | 7325 | 47.4167 | 10.9794 | 2650.0 |

**Table A3.** Error measures of SARAH-2 validation for all cases up to 465 min of forecast time. These values were calculated for the area of Europe. The upper row of each date represents the relative absolute bias in % and the lower one shows the absolute bias in $W/m^2$ respectively.

| Date | Forecast Time (Min) | | | | | | | |
|---|---|---|---|---|---|---|---|---|
| | **45** | **105** | **165** | **225** | **285** | **345** | **405** | **465** |
| 7 August 2017 | 9.10 | 11.52 | 13.62 | 15.28 | 17.08 | 17.77 | 19.67 | 23.43 |
| | 58.64 | 79.25 | 93.11 | 97.01 | 93.38 | 79.13 | 61.58 | 40.78 |
| 11 August 2017 | 11.52 | 14.32 | 16.26 | 17.70 | 19.36 | 20.76 | 23.32 | 27.17 |
| | 66.73 | 89.09 | 101.11 | 102.40 | 95.17 | 82.92 | 64.72 | 59.63 |
| 15 August 2017 | 8.98 | 11.93 | 13.59 | 15.73 | 17.60 | 18.04 | 19.86 | 23.04 |
| | 58.41 | 82.36 | 92.62 | 98.72 | 94.24 | 77.21 | 58.65 | 54.92 |
| 28 August 2017 | 11.70 | 16.25 | 19.48 | 21.93 | 24.44 | 26.46 | 28.83 | 32.95 |
| | 66.24 | 96.75 | 113.62 | 117.03 | 108.83 | 91.3 | 92.82 | 51.57 |
| 29 August 2017 | 10.59 | 13.58 | 16.20 | 17.64 | 19.67 | 21.69 | 24.21 | 30.17 |
| | 60.23 | 81.35 | 95.03 | 94.51 | 87.79 | 73.67 | 51.14 | 47.41 |
| 1 September 2017 | 10.47 | 14.91 | 17.00 | 18.51 | 20.21 | 20.84 | 22.39 | 23.86 |
| | 59.55 | 90.18 | 101.61 | 101.11 | 92.97 | 75.20 | 52.62 | 31.11 |
| 7 September 2017 | 14.60 | 20.20 | 23.52 | 24.48 | 26.33 | 27.21 | 28.16 | 25.55 |
| | 65.20 | 94.47 | 107.60 | 102.78 | 91.44 | 73.71 | 46.93 | 22.76 |
| 17 September 2017 | 12.59 | 18.04 | 21.43 | 22.63 | 23.31 | 23.77 | 25.32 | 26.72 |
| | 59.18 | 88.51 | 102.30 | 98.81 | 83.44 | 63.65 | 38.11 | 22.92 |
| 19 September 2017 | 14.26 | 19.87 | 23.15 | 24.63 | 26.12 | 26.36 | 26.77 | 23.74 |
| | 64.52 | 94.35 | 106.79 | 103.11 | 88.90 | 66.95 | 38.74 | 20.08 |
| 22 September 2017 | 11.67 | 15.23 | - | 19.47 | 21.14 | 23.48 | 24.47 | 26.24 |
| | 52.09 | 71.82 | - | 80.67 | 70.85 | 57.53 | 33.92 | 20.08 |
| 26 September 2017 | 15.13 | 19.79 | 22.65 | 24.32 | 26.26 | 27.90 | 27.83 | 25.71 |
| | 64.69 | 89.15 | 98.55 | 96.67 | 83.70 | 63.01 | 35.77 | 17.62 |
| 30 September 2017 | 12.79 | 17.30 | 20.62 | 22.14 | 25.03 | 26.79 | 31.07 | 31.91 |
| | 53.20 | 74.10 | 82.72 | 78.56 | 67.87 | 47.80 | 29.88 | 15.29 |
| 1 October 2017 | 16.76 | 21.20 | 23.70 | 25.71 | 28.12 | 32.62 | 38.52 | 37.42 |
| | 65.97 | 87.08 | 91.81 | 87.65 | 73.45 | 57.72 | 37.50 | 18.60 |
| 2 October 2017 | 18.10 | 23.08 | 26.33 | 28.05 | 30.56 | 32.27 | 35.01 | 35.59 |
| | 73.02 | 97.73 | 105.55 | 99.08 | 81.54 | 59.67 | 38.27 | 19.53 |
| 3 October 2017 | 16.49 | 20.51 | 22.96 | 24.58 | 26.91 | 28.97 | 30.11 | 31.52 |
| | 67.94 | 89.16 | 95.75 | 92.08 | 78.98 | 57.96 | 35.44 | 17.39 |
| 4 October 2017 | 16.46 | 19.63 | 21.55 | 22.70 | 23.71 | 25.37 | 25.80 | 25.83 |
| | 66.59 | 84.74 | 87.86 | 82.82 | 67.34 | 48.45 | 29.22 | 13.76 |
| 7 October 2017 | 17.26 | 20.26 | 21.60 | 22.52 | 23.32 | 24.32 | 23.22 | 22.59 |
| | 63.81 | 79.09 | 80.57 | 77.10 | 62.80 | 44.35 | 26.26 | 11.87 |

**Table A4.** Error measures of SARAH-2 validation for all cases up to 465 min of forecast time. These values were calculated for the area of Europe. The upper row for each date represents the relative RMSE in % and the lower one shows the RMSE in W/m$^2$ respectively.

| Date | Forecast Time (Min) | | | | | | | |
|---|---|---|---|---|---|---|---|---|
| | 45 | 105 | 165 | 225 | 285 | 345 | 405 | 465 |
| 7 August 2017 | 8.59 | 15.07 | 20.71 | 23.51 | 26.39 | 27.54 | 30.82 | 37.92 |
| | 83.88 | 117.61 | 140.33 | 147.39 | 142.00 | 120.23 | 94.27 | 64.14 |
| 11 August 2017 | 17.66 | 22.45 | 25.29 | 27.17 | 29.80 | 31.91 | 35.54 | 41.20 |
| | 101.97 | 138.38 | 154.88 | 153.84 | 142.45 | 123.20 | 94.82 | 59.63 |
| 15 August 2017 | 16.62 | 20.29 | 22.98 | 24.67 | 27.92 | 28.90 | 31.51 | 36.30 |
| | 89.69 | 128.84 | 143.23 | 151.98 | 145.85 | 120.17 | 89.71 | 54.92 |
| 28 August 2017 | 17.19 | 24.15 | 28.91 | 32.73 | 24.44 | 39.61 | 44.28 | 52.94 |
| | 96.96 | 142.22 | 165.55 | 170.24 | 157.76 | 131.24 | 92.82 | 51.57 |
| 29 August 2017 | 16.17 | 20.99 | 25.42 | 28.22 | 31.73 | 35.53 | 42.24 | 53.14 |
| | 91.63 | 124.27 | 146.14 | 146.92 | 136.38 | 115.26 | 82.46 | 47.41 |
| 1 September 2017 | 16.09 | 23.25 | 26.42 | 28.82 | 31.33 | 31.70 | 33.50 | 35.64 |
| | 91.46 | 140.16 | 156.92 | 156.03 | 142.43 | 112.87 | 77.40 | 41.14 |
| 7 September 2017 | 21.42 | 29.39 | 33.93 | 35.58 | 38.06 | 38.21 | 39.12 | 36.04 |
| | 95.26 | 135.83 | 152.04 | 144.92 | 127.03 | 98.48 | 61.37 | 25.36 |
| 17 September 2017 | 20.03 | 28.11 | 32.94 | 34.00 | 34.68 | 34.54 | 37.74 | 38.04 |
| | 94.01 | 137.29 | 156.01 | 146.83 | 122.36 | 90.85 | 55.61 | 23.88 |
| 19 September 2017 | 21.35 | 29.26 | 33.55 | 35.35 | 37.79 | 37.46 | 38.94 | 32.85 |
| | 96.31 | 137.75 | 152.47 | 144.86 | 123.69 | 91.85 | 53.48 | 19.26 |
| 22 September 2017 | 17.59 | 23.31 | - | 29.91 | 32.47 | 35.32 | 38.31 | 37.71 |
| | 78.26 | 108.74 | - | 120.81 | 105.42 | 83.23 | 49.92 | 19.45 |
| 26 September 2017 | 21.47 | 28.35 | 32.46 | 34.35 | 36.85 | 38.65 | 39.78 | 35.48 |
| | 91.53 | 126.61 | 139.11 | 133.64 | 114.23 | 84.33 | 46.76 | 15.44 |
| 30 September 2017 | 18.72 | 25.98 | 30.79 | 32.50 | 36.57 | 40.17 | 47.02 | 43.47 |
| | 77.60 | 110.16 | 121.39 | 112.39 | 95.85 | 68.69 | 38.54 | 11.15 |
| 1 October 2017 | 22.15 | 29.35 | 33.22 | 35.94 | 39.22 | 45.75 | 57.08 | 58.69 |
| | 86.68 | 118.58 | 125.20 | 117.78 | 97.28 | 75.99 | 46.10 | 14.51 |
| 2 October 2017 | 24.11 | 31.23 | 35.61 | 37.91 | 41.33 | 44.22 | 48.51 | 45.29 |
| | 96.87 | 130.68 | 139.94 | 130.20 | 106.39 | 78.32 | 43.98 | 12.90 |
| 3 October 2017 | 22.61 | 28.89 | 32.40 | 34.27 | 37.21 | 40.00 | 42.31 | 43.30 |
| | 92.80 | 123.99 | 132.34 | 124.65 | 105.10 | 76.32 | 40.53 | 12.11 |
| 4 October 2017 | 22.87 | 28.26 | 30.59 | 31.55 | 32.55 | 34.27 | 35.76 | 35.71 |
| | 92.29 | 119.76 | 122.95 | 112.75 | 89.92 | 63.21 | 33.25 | 9.84 |
| 7 October 2017 | 25.41 | 30.05 | 31.78 | 32.05 | 32.13 | 32.39 | 31.94 | 28.01 |
| | 92.53 | 115.66 | 115.80 | 106.17 | 82.92 | 56.12 | 28.07 | 6.67 |

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
