# Peer review of "The Seamless Solar Radiation (SESORA) Forecast for Solar Surface Irradiance—Method and Validation"

_remotesensing, doi:10.3390/rs11212576_

Round 1

Reviewer 1 Report

This study presents a short-term forecasting technique for the effective cloud albedo and the subsequent surface solar radiation using an open source software. All results were validated against satellite-based data and ground-based measurements. It is a well written paper which could be published in the Remote Sensing journal after the following revisions and additional analysis:

Page 1, line 24: Provide references and explain why the 4-hour time horizon fulfill the load flow requirements. An explanation of the SESORA abbreviation is recommended. Page 2, lines 72-74: Provide a reference. Tables 1 and 2: Add the elevation of the stations. Figure 1: Add an image showing the vectors from the two subsequent images. Page 7, lines 163-164: Why do you examined these 17 cases in 3 months and not the same number of cases in a year period as to cover a representative sample of seasonal cloud patterns? The reliability of the proposed nowcast method has to be tested during all seasons. Page 7, lines 166-167: Provide a reference proving this claim or just test cases from different seasons of the year. The patterns are not identical following the variable atmospheric circulation during a whole year period. Page 7, line 189: Try to avoid the use of non-scientific allegations. Page 9, lines 193-195: This is a very interesting perspective for further study. Especially for those PVs that are placed in the black areas. Maybe a quick solution for an operational use of the proposed methodology is to apply the nowcast for a larger area and then use only the displayed (without black areas) after the targeted time horizon. Figure 4: Make the fonts larger and add grid lines. Page 10, lines 225-226: It is recommended to make an additional plot (similar to Figure 4) showing the results only for the effective cloud albedo. This new plot will show the real forecast error without the sun set effect. Page 10, lines 229-231: Provide additional information for the readers. Figure 5: Add the density dimension of values as to highlight the plume’s main behavior. Pages 12-13, lines 284-292: These are just two effects explaining the difficulty of comparison against ground-based measurements. The authors need to explain as well the effect of all potential atmospheric inputs. They now use clouds as an input to SPECMAGIC NOW, but how they handled the effect of the particulate matter, the water vapor, the ground elevation, the cloud phase (ice or water), the surface albedo etc? Are there forecasts for these parameters as to take them into account at the simulations and hence in the comparison against ground measurements? Provide also references. Figure 8: Make the fonts larger and add grid lines. Section 4: Provide a table with the comparable results from the literature together with the finding of this study. This will be a valuable information for the electricity handling entities.

The whole approach merits publication since the results are interesting for the readers and potentially valuable for the energy transmission and distribution system operators. After the proposed corrections the paper will be substantially upgraded and it could be published in the Remote Sensing journal.

Reviewer 2 Report

This manuscript is dedicated to an important topic of the solar surface irradiance forecast, which is needed for a correct planning of grid usage and estimating the available resources on an hour-to-hour basis. The authors suggest and test a forecast scheme, which is based on extrapolation of satellite images obtained from geostationary satellites.

The manuscript is topical, is well written and logically organized, and I believe it can be published in the journal after minor revision, which should address both general and minor comments below.

General comments

Methodologically, the approach is not new. Lagrangian trajectories method has been used for years to predict oceanic and atmospheric movements and it’s not clear why two point extrapolation should be any better than this well-established method. I do not suggest to rewrite the codes and to test the Lagrangian scheme, but the manuscript shall contain at least a section comparing the differences between the suggested methodology and the previous methods which serve similar purposes. Given that the geostationary satellites provide a series of images available for the analysis, it is not clear why the authors limit themselves just to the latest two shots. Indeed, the motion vector needs only two points, but the rotations typical for atmospheric dynamics will be better represented in polar coordinates with a moving center, which will require more than two points. Unless I missed something, I’ve got an impression that the authors have chosen a rather simple model and thoroughly tested and validated it, but the observed biases cannot be removed in the framework of this mode, so they leave the reader with this information, promising the “curved trajectories” only in the conclusions. Perhaps, the manuscript needs a short discussion/back of envelope estimates, which would give an idea of whether the existing biases can be improved. The biases in W/m2 are presented in the same manner both for sun in zenith and for the twilight, but the relative error will be different in these cases. The authors provide these estimates, too, but the discussion and the conclusions lack the background radiance values when the biases are discussed. A bias of 100 W/m2 on top of 1300W/m2 does not mean the same for the grid as the same bias observed for 300 W/m2 radiance.

Minor comments

Lines 11, 12 and elsewhere: does this accuracy make sense? I would say that even 0.1 W/m2 accuracy is too optimistic and 42 W/m2 and 97 W/m2 wouldn’t look odd to me.

Line 25: the manuscript is supposed to provide a general approach overview, but this sentence might make the reader feel that it is more of a regional nature. Perhaps, one can present this information as an example or move elsewhere.

Page 3, Table 2 – this is a long table, which breaks the flow. Moving it to Appendix is not a requirement, though.

Equation 3 – indeed, this equation doesn’t work for optically thick cases. Can it be that the fully covered pixels or the partially filled pixels with CAL > 0.8 are responsible for the biases? Please, comment.

Page 8 – here’s another table which might be moved to the Appendix for the sake of readability.

Line 20 and elsewhere – it is not recommended to use personal form “you can see” in the scientific texts. “One can see” reads smoother and is consistent with general rules.

Figure 5: this representation is not informative because the overlapping points do not allow to estimate the real density. One of possible solutions is to use the approach shown, e.g. in Fig. 12 of Dawkins et al., 2018, DOI: 10.1029/2018JD028742.

Reviewer 3 Report

This paper describes an irradiance forecasting method for forecast horizons between 15 mins and 8 hours (the abstract claims that forecast are made between 15 mins and 4 hours, but that is contradicted in the paper). The forecast is based on an optical flow method applied to two consecutive satellite images. The optical flow forecast defines the “effective cloud albedo” which is transformed to irradiance using the SPECMAGIC NOW method.

I cannot recommend this paper for publication. The main reasons, detailed below, are (i) the paper does not make a clear case for what forecast horizons the proposed method is actually useful for; (ii) the validation amounts to showing mean square errors, but a validation against even simple other forecasting is missing, which makes it impossible to evaluate the skill of the proposed method; (iii) the paper does not provide sufficient details to be able to implement even the fundamental ideas that lead to the proposed forecasting scheme.

Detailed criticism.

1. The paper claims to provide a “validated” method. The validation amounts to showing root mean square errors (RMSE). Some of the RMSE values are tainted by the time of day, leading to smaller RMSE towards the end of day (when the sun sets). These RMSE values are not useful (and plots of RMSE without time-of-day correction should not be shown). While the author do account for a time-of-day correction, an essential part of any validation should include a comparison to other forecasting schemes. For short horizons (15 mins to 1 hr), a comparison to a “persistence” or “smart persistence” must be done. For longer horizons (>6 hr), numerical weather models (or even a basic average!) define the baseline forecast skill (but you should also compare to persistence for any forecast horizon). Such a comparison must be included to be able to evaluate how skillful a forecasting scheme, on different time horizons, really is. The authors might discover that their scheme is ideal for intermediate forecast horizons (between persistence and NWP models). They might also discover that their scheme is useless for any forecast horizon.

The authors also tend to evaluate forecast skill only on relatively long horizons (several hours), but claim that their scheme is useful for 15 mins - 4 hrs (see abstract). This claim is not supported by the numerics shown and, hence, the presentation is not suitable for publication.

I find the linear regression plots (see, e.g., Fig. 5) not convincing for longer horizons (the regression does *not* suggest that this method is useful).

2. Boundary conditions are not treated adequately. The authors briefly mention that the area over which their forecast is useful shrinks over time, due to the inadequate treatment of the boundaries, but no remedy is suggested. The authors will find a useful solution to this problem in Harty et al., Solar Energy 185, 270-282 (2019).

3. Insufficient detail on essentials of the method. Optical flow works by deriving a velocity field from two consecutive satellite images. The authors fail to report the time interval between two consecutive satellite images. This is essential for being able to understand how well and even why the method works (or not). If the interval is short, the velocities are likely to be very useful for short-term forecasts; if the time interval is long, the entire approach is questionable. Again, I wonder what the purpose of the proposed scheme is, because it is common sense that optical flow is only useful over relatively short forecast horizons of mins-1 hr, not 4-8 hrs as suggested in the paper.

4. Important issues are not discussed. The optical flow method described here only provides an "effective cloud albedo". A conversion of this quantity into irradiance on the ground requires that the cloud height and the relative angle of the sun are known. The paper does not provide any indication of how these issues are dealt with (other than referring to SPECMAGIC NOW), which makes it difficult to evaluate the adequacy of the proposed scheme for the proposed task.

5. Optical flow induces divergence. In my own experience with optical flow and short-term forecasts of irradiance, I found that an important issue is that an optical flow scheme introduces unphysical divergence into the effective cloud albedo field (again, see Harty et al., Solar Energy 185, 270-282 (2019)). This issue is not discussed in this paper and I must wonder why. In my experience, simply removing divergence from an optical flow forecasts leads to much improved results. Please include a discussion of divergence within the cloud albedo in any future revision.

Round 2

Reviewer 1 Report

The paper was significantly improved and now can be published in the Remote Sensing journal.

Author Response

Thank you very much for taking your time to evaluate our manuscript.

Reviewer 3 Report

I thank the authors for addressing many of my concerns. 

I want to clarify a few things.

I do not object to using RMSE, but my point is that just reporting RMSE can not be considered validating a model. Figure 9 helps, but I would also like to see how RMSE/forecast skill behaves as a function of the forecasting time. I believe the skill of the optical flow method will degrade with the forecasting time and it would be interesting to see how quickly and how the rate with which skill degrades compares to other methods. I would further suggest to always correct the RMSE for the time of day -- it is clear that RMSE towards the end of the day is dominated by the sun setting, hence all methods should perform equally at that time. Similarly, I do not object to using linear regression plots, as in Figure 5. I conclude from Fig.9, however, that the method performs rather poorly. The authors come to the opposite conclusion and I wonder why. About cloud height: there must be a misunderstanding between the authors and myself. I attribute this to the manuscript being difficult to read unless one is familiar with the authors previous work and a masters thesis. For example, in the abstract the authors state "the optical flow of the effective cloud albedo" (please remove repeated "the"). I understand the method to be like this: the optical flow translates effective cloud albedo and SPECMAGIC is used to compute irradiances. If I want to compute irradiances *on the groud* I need to know the cloud height and the solar zenith angle. It would be good if the authors could clarify this point. I am also unsure about how precisely the method works. I assume that two satellite images (with 15 min in between) are collected in the morning. An optical flow is computed and then applied to the later of the two images. Is this correct? This means that a 4h forecast is based on a satellite image from 4h ago? Is this procedure repeated every 4 hrs? And if so, what forecast horizon was used to generate the RMSE numbers mentioned in the abstract? Divergence: I am glad to learn that TV-L1 (an optical flow method?) considers divergence. Again, confusion about divergence and many related issues could have been avoided if there was more information in the paper so that it can be read and understood independently of previous work by the authors.
